# Progress on Research Regarding Ecology and Biodiversity of Coastal Fisheries and Nektonic Species and Their Habitats within Coastal Landscapes

Mark S. Peterson * and Michael J. Andres

Division of Coastal Sciences, School of Ocean Science and Engineering, The University of Southern Mississippi, Ocean Springs, MS 39564, USA; michael.andres@usm.edu
* Correspondence: mark.peterson@usm.edu

**Abstract:** This paper aims to highlight the new research and significant advances in our understanding of links between coastal habitat quality/quantity/diversity and the diversity of fisheries species and other mobile aquatic species (hereafter nekton) that use them within coastal landscapes. This topic is quite diverse owing to the myriad of habitat types found in coastal marine waters and the variety of life history strategies fisheries species and nekton use in these environments. Thus, we focus our review on five selective but relevant topics, habitat templates, essential fish habitat, habitat mosaics/habitat connectivity, transitory/ephemeral habitat, and the emerging/maturing approaches to the study of fish-habitat systems as a roadmap to its development. We have highlighted selected important contributions in the progress made on each topic to better identify and quantify landscape scale interactions between living biota and structured habitats set within a dynamic landscape.

**Keywords:** review; estuary; connectivity; nekton; techniques

Individual coastal and marine systems are components of the coastal aquatic landscape and as such, organisms that use these landscapes during a portion of their life history must, by definition, encounter a number of "environments" and "habitats" [1]. These mosaics are a mix of interconnected vegetated, lithogenous, and human- and animal-made structures that ultimately act as templates on which population and community dynamics occur [2–4]. Thus, effective conservation and management of coastal ecosystems must take into account both the variability in abiotic conditions and the nested structural habitat component [5,6]. However, many linkages between habitat and fisheries and nektonic species production have been and continue to be altered by urbanization. Thus, we are studying these linkages (functions) for sustainability and biodiversity while they are changing in quality, quantity, and interconnectedness [7–9]. Another important consideration is that despite our best intentions with restoration activities, we are ultimately exchanging one habitat type for another and restoring habitats in spatially different locations within the landscape, which further alters the community dynamics and linkages.

## 1. Habitat Templates

In an often overlooked study, Ryder and Kerr [1] identified the need to set habitat structure within the appropriate, but dynamic, abiotic environmental condition when restoring habitat. Although their argument was focused on salmon stocks, it can be used across a spectrum of ecosystems as it is hierarchical in nature. Thus, set within an abiotic framework, habitat templates and the ever-changing nature of coastal landscapes establish and maintain the underpinning for coastal biodiversity. Ryder and Kerr [1] have defined environment as the "… total physical, chemical, and biological surroundings of an organism, including habitat and other organisms." "They view environment and habitat hierarchically as"… the pervasiveness of relatively structureless environment which provides background ambience, against the localized and highly structured habitat which acts as a center of

organization and attractor for fish (and nekton) communities." McCoy and Bell [10] and Hoss and Thayer [11] supported this approach. In their review of estuarine systems, Simenstad et al. [12] indicated "...habitat structure at the ecosystem scale encompasses the configuration, and arrangement and connectivity, of habitat elements that typify most estuaries". Based on their review, they suggested eight major steps to enhance a fuller understanding of habitat-biotic linkages ranging from (1) coastal habitat delineation relative to production and food web processes to (2) use of long-term data sets on habitat variability, and (3) comparative studies on habitat function along with five other estuarine-specific topics [12]. Some of which are addressed in this Special Issue volume.

## 2. Early Papers on Habitat Use and Nursery Function

Early studies on nursery habitat use and function were simplistic, but they provided a platform whereby others could focus their collective efforts with extensive and detailed field studies. These subsequent studies incorporated not only biotic collections but also abiotic factors known to be important for the survival, growth, and other life history vital metrics. Joseph [13] identified three broad criteria that must be met if an area is to serve a significant nursery role. These criteria are: "1. The area must be physiologically suitable in terms of chemical and physical features; 2. it must provide an abundant, suitable food supply with a minimum of competition at critical trophic levels; and 3. it must in some way provide a degree of protection from predation." Although these were based on a vast array of long-term data sets and are overly simplistic, they were fundamental to subsequent studies. Two of the earliest and far-reaching landscape-scale studies of fisheries and nektonic species and habitat occurred in the Cape Fear River estuary, North Carolina, USA, by Weinstein [14] and Weinstein et al. [15]. There were a number of key results of these two landscape-scale studies. First, multivariate analyses demonstrated that each saltmarsh complex was unique. Second, assemblage differences were correlated to both salinity gradients and an "edge effect" where saltmarshes closest to the river mouth were species rich due to seasonal invasion by low densities of reef, nearshore, and shelf marine species coupled with freshwater fishes in brackish marshes during high flow periods. Finally, these data sets indicated ocean-spawned species were exported to the adult offshore marine habitat annually in the form of living biomass.

Peters and Cross [16] further outlined habitat types and other habitat structural features (e.g., habitat edges, river plumes, turbidity maxima, etc.) not considered in earlier studies as important to fisheries species and nekton in coastal and marine ecosystems. They relied heavily on Ryder and Kerr [1] in terms of abiotic factors and habitat structure hierarchically in defining fish habitat and thus nurseries. They concluded that although we know a lot about fish and habitat independently, ecologists and managers have yet to examine quantitatively fish-habitat interactions for a single species let alone for a single (or multiple) species across a mosaic of habitat types within a landscape. This early hierarchical concept was further formalized in Hoss and Thayer [11], who discussed the importance of studying estuarine and coastal nearshore marine fish habitat connectivity patterns instead of individual habitat types. Finally, in a review of the impacts of fishing pressure on fisheries species and their habitat, Langston and Auster [17] indicated the need to " . . . develop a predictive capability given a particular management protocol so that fishery management becomes tactical and strategic rather than anecdotal and speculative." They postulated that to achieve this, ecological processes must be quantified and then applied that will allow the maintenance of habitat integrity (and therefore biodiversity) across the interconnected landscapes of coastal and marine ecosystems.

## 3. Essential Fish Habitat

Beck et al. [18,19] reviewed, focused, and delineated quantitatively essential fish habitat (EFH) requirements and approaches to better define it in a multitude of habitat types. Following Beck et al. [18,19], a habitat is considered a juvenile nursery if its contribution per unit area to the production of individuals that recruit to adult populations is greater,

on average, than production from other habitats in which juveniles occur. One vital link needed to develop a complete understanding of which habitats serve nursery function is quantifying the connectivity among habitat types throughout ontogeny for a species [20]. These data are valuable for resource managers in delineating EFH as well as promoting the conservation of the mosaic of habitat types and biodiversity in coastal and marine ecosystems. Dahlgren et al. [21] refined the initial EFH concept as the Effective Juvenile Habitat (EJH) concept to refer to habitat types that make a greater than average overall contribution to adult populations see [22,23] rebuttals. Regardless of these different schools of thought, typically EFH is associated with structured habitat types like submerged aquatic vegetation (SAV), seagrass, mangrove, coral reefs, or salt marshes [24–28]. However, this is not always the case e.g., [11,29], as many pelagic nekton species like menhaden, silversides, and anchovies comprise a significant biomass component of coastal and marine environments and appear not to require discrete habitat structure. Maintaining the availability and connectivity of the mosaic of habitat types is critical to a diverse ecosystem [30,31].

## 4. Habitat Mosaics/Habitat Connectivity

Aquatic resource conservation and sustained fisheries species and nekton production requires a combined approach of managing the mosaic of habitat types used by living biota within a particular landscape coupled with management of fishing effort for commercial/recreational species e.g., [5,17]. Sheaves [32] in his review of connectivity indicated the multifaceted linkages among the diverse habitat types comprising ecosystem complexes like the coastal ecosystem mosaic (CEM)—the tightly interlinked coastal, estuarine, wetland, and freshwater habitats at the interface of land and sea. The diversity of connected habitat types is integral to an array of organism life history vital metrics, with connectivity between habitat types being crucial to important functions like nursery use.

Although debatable, it is generally believed that aquatic organisms survival depends upon approaching a physiological optimal range of conditions first and then behaviorally searching out the appropriate life-stage-dependent habitat. Recently, Fulford et al. [33] extended a terrestrial model of small-scale movement patterns to describe habitat choices of an index juvenile estuarine fish using the critical laboratory experiments. They found movement was influenced by both spatial and temporal patterns in habitat quality, and it was a balanced mix of both that resulted in an optimal juvenile distribution. Model outcomes indicated a hierarchical approach is best for describing habitat choice in juvenile fishes. However, there are tradeoffs among available factors that determine the distribution of animals, such as the presence of suitable food [34,35], structural complexity [36], the proximity to linked habitat types [37], transport of planktonic propagules [38], and recruitment by opportunistic species (i.e., freshwater or marine). As Able et al. [39] noted, based on long-term saltmarsh nekton collections in Delaware Bay (USA), it is clear that there are few truly marsh-dependent species as almost all species simultaneously use a variety of other bay and river habitats that vary seasonally and ontogenetically.

Sheaves et al. [22] recognized that nursery ground value cannot be measured solely as a numeric contribution to the adult stock e.g., [18,21], but must include the contribution to future generations. They argued that preserving "keystone" habitat types at the expense of other habitat types is philosophically problematic because coastal and marine ecosystem are linked in space and time and the degradation of non-keystone habitat types can lead to alteration of these important landscape linkages for a number of species we wish to protect. Thus, because of our current lack of understanding of these linkages and which habitat types may be more "valuable and worth saving," decisions made relative to these presumed more valuable habitat types alone may likely produce unknown consequences based on these choices. For example, Sambrook et al. [40] reviewed the use of non-coral reef habitat types by fishes and for the 170 species with complete life history data, about 76% used non-reef habitats in juvenile and adult life stages. This use of non-reef habitats by "coral reef" fishes was widespread, indicating resource managers need to consider broader scales and different forms of connectivity when dealing with human induced impacts.

Finally, urbanization, climate change, and subsequent restoration in real-time are changing many of these linked relationships between environment-habitat processes in coastal and marine mosaics [8,9,41].

Moreover, Sheaves et al. [42] argued the true value of coastal nurseries for nekton is much more extensive and requires other fundamentally important ecosystem processes. Not considering these broader aspects may allow for "suboptimal conservation outcomes," especially given the increasing urbanization of coastal and marine systems and the likelihood that protection will be focused on specific locations, (see [43] on marine spatial planning) rather than landscape habitat mosaics. Finally, in a recent meta-analysis of nursery function, Lefcheck et al. [30] confirmed the basic nursery function of certain structured habitats, which lends further support to their conservation, restoration, and management at a time when our coastal environments are becoming increasingly urbanized (e.g., [42,43]). This continued need for further long-term and location or habitat-specific studies are especially critical for nekton as are the studies of the functional significance of estuarine-specific contributions to continental shelf metapopulations [44]. Lefcheck et al. [30] advocates for a renewed emphasis on more direct assessments of juvenile growth, survival, reproduction, and recruitment compared to increasingly complex approaches that examine nursery function in a landscape.

## 5. Transitory/Ephemeral Habitat

We believe a special statement related to non-structural transitory and ephemeral structured habitat types is necessary because these habitats are typically often not examined as closely as traditional structured habitat types e.g., [11]. Transitory/ephemeral habitat types appear seasonally or aperiodically within a coastal landscape and enhance biodiversity of organisms within the landscape. Ephemeral structured habitats (e.g., drift algae, Sargassum, mobile macrophytes, bryozoan mats) and non-structured habitats [5,11] (e.g., local hydrodynamics, upwelling zones, coastal currents, etc.) appear to provide organisms within a landscape additional foraging sites and predator refuge indicating these could be a valuable, temporary habitat type for small, motile species or early life stages of ecological/commercial/recreational species [45–47]. Studies on various drifting ephemeral habitats indicate that species richness and/or biomass of living biota are highly correlated with these ephemeral habitats e.g., [48–51], likely due to increased complexity and living space that may decrease intraspecific competition and predator encounter rates. These ephemeral habitat types clearly enhance biodiversity in coastal and marine environments, albeit, on reduced temporal and/or spatial scales. Furthermore, ephemeral habitats and rafting of such habitats can increase dispersal and population connectivity for coastal species [52].

## 6. New Approaches/Techniques

Like all fields, the study of coastal fisheries species and other nekton have benefited greatly from technological advancements and increases in computing power. Adequate sampling of coastal systems is difficult, in part because of the rate of change (both natural and anthropogenic) in these environments, how connected these systems are to other habitats or ecosystems (e.g., terrestrial or adjacent systems), and how fisheries and nektonic species use the coastal realm through their ontogeny. Traditional sampling methodologies (physical capture) all have inherent selectivity biases as well which make additional techniques necessary to adequately understand and quantify these species-habitat linkages. Here we highlight just a few of these techniques that continue to refine and reshape our understanding of connectivity and biodiversity patterns in the coastal zones. This list is not meant to be exhaustive by any means and should not be taken as such.

The microchemistry of hard parts (e.g., otoliths, spines, eye lenses), although not necessarily a new approach, has been instrumental in determining habitat use, population connectivity, natal origin, and migrations of various fish species [53]. This line of research has even moved into understanding ecological disturbances like hypoxic events [54],

fisheries management [55], and efficacy of marine protected areas [56]. This technique is especially valuable for understanding the connectivity and potential for dispersal of larvae when combined with genetics approaches as data pertaining to space (microchemistry), time (otolith rings), and population (genetics) are all integrated.

Similar to microchemical analyses, genetic analyses are not novel; however, the sustained decrease in cost of sequencing, the optimization of PCR, and sequencing techniques, and the growing sequence library for coastal species has allowed for the field of environmental DNA (eDNA) to flourish. eDNA studies have some advantages over traditional sampling techniques (albeit with some caveats) in that they are generally less expensive, require less field time, and are less invasive. This field has provided novel insight into range determination for rare/protected species [57,58], detection of non-natives [59], and has opened the door for metabarcoding studies to roughly characterize the biodiversity of various habitats with a single water sample [60,61]. However, caution must be noted in that sequence libraries (barcodes) are still incomplete for most nekton species, metadata standards for sequence data are not well unified, and incomplete vouchering of taxa used to derive sequence data is still common [62]. This is especially important as ranges for many taxa are being altered in the face of a changing climate; therefore, emphasis on training future scientists with a strong coupling of taxonomic and molecular expertise is still needed.

The invention of inexpensive, small cameras for action sports have been repurposed by coastal ecologists for developing a more complete understanding of shallow, coastal habitats. Remote underwater camera station (RUVS) arrays for reef fish surveys and baited RUVS (BRUVS) for large coastal predators and deep water applications have a longer history of use [63,64], but the cost of such equipment and size largely precluded their use in coastal waters. The smaller "GoPro" style cameras have the ability to function in a passive or baited way to document fine-scale habitat use, community assemblage, behavioral and feeding patterns that would otherwise prove difficult using traditional sampling gear [65–67]. These systems are particularly effective in habitats like mangroves and intertidal oyster reefs where water clarity is relatively clear, but effective net sampling is difficult because of the complex shape and silty/detrital nature of the sediments. These systems are limited in breadth of habitat sampled, deployment time, and memory card/battery life, but are especially effective when combined with other concurrent sampling techniques. An extension of the decrease in size and increase in sensor resolution of small cameras is the increasing use of drone (or unmanned aircraft systems) technology for surveying difficult to access coastal habitats. Drones have been successful in reducing the cost of traditional aerial surveys for mapping fish nursery grounds [67], estimating the number individuals in a spawning aggregations [68] and for determining the extent of intertidal and subsurface habitat like coral reefs [69] and oyster reefs [70]. This application has some current limitations related to weather conditions suitable for flying, local governance for airspace, and water clarity/surface reflection, but the last can be partially overcome because of the large number of sensors available for mounting on the drones. Recently, mapping subtidal estuarine habitats using a remotely operated underwater vehicle (ROV) piloted from a boat was developed and tested in a South African estuary. Similarly, ROVs have seen a similar reduction in size/cost, that has enabled their application in shallow water and have shown promise for surveying/mapping submerged aquatic vegetation beds [71].

## 7. Summary

Our understanding of nuances associated with the ecology and biodiversity of fisheries species and other nekton using the mosaic of habitat types in the coastal zone has improved and new technologies have helped address many questions that will allow us to focus our hypotheses concerning fish–habitat relationships. However, the dynamic nature of these habitats coupled with the ever-increasing modifications of the coastal zone (in the name of both development and restoration) and a warming climate mean that new research

trajectories are always occurring. Restoration projects within these mosaics require clear goals for success as well as continued monitoring to determine if goals are being met and to document any unintended consequences of restoration activities. Other aspects that require additional data collection to completely understand the diversity and function are related to the role parasites and pathogens have on the ecology of nekton in these systems. Parasites are often ignored by researchers of free living organisms despite their ability to act on the individual, population, and community level [72], only becoming a focus once a given problem raises significant concern [73]. The role of non-fisheries nekton in coastal mosaics are especially important as well. Recent works have further demonstrated the need to understand the ecology and diversity of cryptobenthic fishes in many coastal systems including coral reefs [74,75], soft sediments [76], and estuarine habitats [77–79]. Although there has been significant strides in the understanding of living biota-habitat connectivity over the last three decades, more work has to be done. Future research needs to be completed in a manner that will allow resource managers to assess the continued loss of habitat and fisheries and nektonic species collectively. This research is fundamental to better quantify the functionality, sustainability, and thus the long-term biodiversity of these economically and aesthetically valuable ecosystems and to assure their contribution to a viable planet.

**Author Contributions:** Conceptualization M.S.P. and M.J.A.; methodology, M.S.P. and M.J.A.; writing—original draft preparation, M.S.P. and M.J.A.; writing—review and editing, M.S.P. and M.J.A. All authors have read and agreed to the published version of the manuscript.

**Funding:** This research received no direct external funding.

**Institutional Review Board Statement:** Not applicable.

**Informed Consent Statement:** Not applicable.

**Data Availability Statement:** No new data were created or analyzed in this study. Data sharing is not applicable to this article.

**Acknowledgments:** We thank numerous undergraduate and graduate student researchers whose work inspired us and ultimately led to the development of this Special Issue and to write this paper.

**Conflicts of Interest:** The authors declare no conflict of interest.

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
