# Peer review of "Progress on Research Regarding Ecology and Biodiversity of Coastal Fisheries and Nektonic Species and Their Habitats within Coastal Landscapes"

_diversity, doi:10.3390/d13040168_

Round 1

Reviewer 1 Report

Summary:

This manuscript briefly reviews the ecology and biodiversity of fisheries with respect to habitat for coastal and estuarine systems. The manuscript succinctly summarizes the recent history of coastal habitat studies and describes a series of concepts driving fish-habitat relationships, including hierarchical habitat definitions, variation in ontogenetic habitat use and value, and an emphasis on linkages and connectivity across habitat types.

This review appears to serve as an introduction to a special issue devoted to the topic of understanding links between coastal habitat and fisheries species. More consistency in terminology (e.g., coastal/marine/estuarine, fisheries species/nekton) and more clear organization would clarify the review for the reader and would make the review a more effective introduction for the rest of the Special Issue. Examples and suggestions for organization and terminology are below. I recommend that this review be revised to more clearly introduce the Special Issue topic and address the emphases of the title.

Major comments from this reviewer are limited to organization and topic inclusion. Major and minor comments are included below and summarized by line number where possible.

Major Comments:

Organization: It is unclear to the reader how the subheadings relate to the introductory paragraph and why they were chosen. Modifying the introduction to better introduce subsequent sections, both in further justifying the inclusion of these topics and as a roadmap for the reader to follow, would be helpful. This also would help the review serve as an introduction for the rest of the Special Issue. For example, Section 1 defines habitat but without context it is unclear what, if anything, the reader should take away from that section.

Terminology: The paper uses multiple terms interchangeably, making it difficult for the reader to identify the specific focus of the review. First, the paper jumps between the broad description of coastal habitat/ecology and the specific of estuarine habitat/ecology. By itself this isn’t an issue, however given the manuscript title I expected a more consistent emphasis on estuaries. Presumably the authors are aware of at least some of the articles in the Special Issue and can therefore emphasize habitats appropriately. I suggest reframing the manuscript to be consistent in its treatment of this distinction (effectively done in Lines 159-162) as best suits the needs of the Special Issue.

Second, the title refers to fisheries species, suggesting that the emphasis of the review is on recreationally or commercially important taxa as opposed to general coastal/marine biodiversity. However, this nuance is not addressed consistently throughout the manuscript. I suggest either focusing more explicitly on fisheries-specific examples (e.g., Beck et al. 2001) or adjusting the scope of the review to highlight coastal/marine biodiversity more generally. For example, management of fishing effort is mentioned in line 111, but elaboration is missing. In the summary the author’s highlight the importance of non-fisheries nekton, but the majority of the review is not fisheries-specific. Even if intentional, explicitly clarifying early in the introduction will help the reader.

Topic Inclusion: This ‘mini-review’ is brief, and as such can not include all pertinent literature. However, this manuscript would benefit from more explicitly acknowledging certain topics:

  • The importance of habitat types other than ‘structured’ elements. This is particularly true for estuaries, where river plumes, turbidity maxima, and local hydrodynamics create temporary or permanent features which can dramatically alter juvenile retention, community and food web dynamics, nutrient subsidies/retention, and overall habitat suitability. Upwelling zones, coastal currents, etc. similarly impact coastal/marine habitats. This could be added to the section on Transitory/Ephemeral habitat, or wherever else the authors deem appropriate.   
  • Section 1 is devoted to multiple definitions of habitat and habitat templates within coastal/estuarine systems, but it is unclear how various habitat definitions differ, and what they add to the manuscript. I recommend removing lines 43-45, and better contextualizing the Ryder & Kerr and Simenstad et al. definitions within the manuscript. This may include providing an example of the 8 steps identified by Simenstad et al., or some organizational modification.

Minor Comments:

Lines 20-22: Remove the word ‘Unique’, or otherwise rephrase. Are ‘non’ unique systems not components of the landscape? Or do the authors refer to individual systems that are part of the larger landscape?

Line 21-22: Move ‘by definition’ after ‘must’

Line 23: Rephrase. ‘Subsytem’ is unclear, particularly following the use of the word ‘system’ and ‘ecosystem’ in the first sentence.

Line 28-29: It is unclear what is meant by ‘despite habitat…dynamics occur’. Recommend removing the phrase and making the linkage alteration the focus.

Line 34: Add ‘and linkages’

Lines 36-47: It is unclear how various habitat definitions differ, and what they add to the manuscript. Recommend removing lines 43-45, and better contextualizing the Ryder & Kerr and Simenstad et al. definitions within the manuscript.

Line 65: Change to ‘stated that each studied saltmarsh complex was unique’

Line 77: Unclear what is meant by comparing one species to a mosaic of habitat types.

Line 93: Insert ‘needed’ after ‘vital link’

Line 115: Confirm plural of ‘life histories’, rather than ‘life history’

Line 129: Missing bracket around [38]

Line 164: change ‘increasing’ to ‘increasingly’

Line 174: Remove the extra space after ‘commercial/’

Line 200-201: Recommend changing spatial/chronological to nouns rather than adjectives (i.e., space, time)

Line 218: Insert ‘of’ between ‘understanding’ and ‘shallow’

Line 221: Remove ‘been used’

Line 223-224: Rephrase ‘community assemblage, behavioral and feeding studies’ to something like ‘community assemblage, and behavioral and feeding patterns’ for consistency with the rest of the sentence.

Line 235: Remove ‘obviously’

Line 243: Remove ‘certainly’

Line 258-259: Clause ‘Although there…last three decades’ should be rephrased for clarity/brevity.

Lines 261-262: Recommend rephrasing to emphasize that long-term biodiversity is a result of functionality and sustainability.

Line 263: Remove ‘future presence and’

References

General comments: Inconsistent use of semicolons or commas to separate authors

Ref 6, 13: Extra spaces in title.

Author Response

The reviewers comments were spot on and as a editor myself, I appreciate the focus to detail of our text. We hope your agree we made all changes/edits/additions requested to your satisfaction. 

Reviewer 2 Report

I really enjoyed reading this manuscript and feel that it makes a valuable contribution to the literature on coastal marine species and their habitat. The manuscript is well written and I suggest that this manuscript be published after minor revisions. Minor revisions as well as suggestions for papers are below:

Page 2 (lines 62-63): Provide more detail on the studies by Weinstein (1979) and Weinstein et al (1980) for readers not familiar with these papers

Page 2 (line 74): Provide some background to the study by Ryder and Kerr (1989)

Page 5 (lines 219-222) restructure the sentence: “Remote underwater camera station (RUVS) arrays for reef fish surveys and baited RUVS (BRUVS) for large coastal predators and deep water applications have a longer history of use [62,63] been used” to “Remote underwater camera station (RUVS) arrays for reef fish surveys and baited RUVS (BRUVS) for large coastal predators and deep water applications have a longer history of use [62,63].”

Page 5 (lines 217-239): Another new technique to map subtidal estuarine habitats is remotely operated underwater vehicles, see Wasserman et al. (2020) Mapping subtidal estuarine habitats with a remotely operated underwater vehicle (ROV). African Journal of Marine Science 42: https://doi.org/10.2989/1814232X.2020.1731598

The following paper reviews the importance of different vegetated habitats for coastal fishes and should be referred to: Whitfield, A.K. (2017). The role of seagrass meadows, mangrove forests, salt marshes and reed beds as nursery areas and food sources for fishes in estuaries. Reviews in Fish Biology and Fisheries 27, 75–110. DOI 10.1007/s11160-016-9454-x

Author Response

Thanks to this reviewer for assisting us to make this a better piece of work.  Your comments/edits/concerns have been addressed and hope you agree.
